# FAK Signaling in Rhabdomyosarcoma

**DOI:** 10.3390/ijms21228422

**Published:** 2020-11-10

**Authors:** Clara Perrone, Silvia Pomella, Matteo Cassandri, Maria Rita Braghini, Michele Pezzella, Franco Locatelli, Rossella Rota

**Affiliations:** 1Department of Oncohematology, Bambino Gesù Children’s Hospital, IRCCS, 00165 Rome, Italy; clara.perrone@opbg.net (C.P.); silvia.pomella@opbg.net (S.P.); matteo.cassandri@opbg.net (M.C.); michele.pezzella@opbg.net (M.P.); franco.locatelli@opbg.net (F.L.); 2Department of Science, University of Rome ‘Roma Tre’, 00146 Rome, Italy; 3Molecular Genetics of Complex Phenotype Unit, Bambino Gesù Children’s Hospital, IRCCS, 00146 Rome, Italy; mariarita.braghini@opbg.net; 4Department of Pediatrics, Sapienza University of Rome, 00185 Rome, Italy

**Keywords:** rhabdomyosarcoma, FAK, focal adhesion complex, sarcoma, FAK inhibitors, cell migration, cell invasion, kinase, myogenesis

## Abstract

Rhabdomyosarcoma (RMS) is the most common soft tissue sarcoma of children and adolescents. The fusion-positive (FP)-RMS variant expressing chimeric oncoproteins such as PAX3-FOXO1 and PAX7-FOXO1 is at high risk. The fusion negative subgroup, FN-RMS, has a good prognosis when non-metastatic. Despite a multimodal therapeutic approach, FP-RMS and metastatic FN-RMS often show a dismal prognosis with 5-year survival of less than 30%. Therefore, novel targets need to be discovered to develop therapies that halt tumor progression, reducing long-term side effects in young patients. Focal adhesion kinase (FAK) is a non-receptor tyrosine kinase that regulates focal contacts at the cellular edges. It plays a role in cell motility, survival, and proliferation in response to integrin and growth factor receptors’ activation. FAK is often dysregulated in cancer, being upregulated and/or overactivated in several adult and pediatric tumor types. In RMS, both in vitro and preclinical studies point to a role of FAK in tumor cell motility/invasion and proliferation, which is inhibited by FAK inhibitors. In this review, we summarize the data on FAK expression and modulation in RMS. Moreover, we give an overview of the approaches to inhibit FAK in both preclinical and clinical cancer settings.

## 1. Introduction

Rhabdomyosarcoma (RMS) is the most common soft tissue sarcoma of childhood, accounting for about 50% of all the soft tissue sarcomas and 8% of all pediatric cancers [1]. RMS cells derive from mesenchymal precursors mainly committed to myogenic lineage that are unable to differentiate [2,3]. In agreement, RMS cells express master myogenic factors such as MYOD and myogenin (MYOG), used for diagnostic purposes. Pediatric RMS includes two main histological subtypes, namely embryonal and alveolar RMS. About 70% of alveolar RMS harbors chromosomal translocations among which the most represented are t(2;13)(q36;q14) and t(1:13)(p36;q14), which encode for the chimeric oncoproteins PAX3-FOXO1 and PAX7-FOXO1, respectively [4], and are indicated as fusion-positive (FP)-RMS. PAX3-FOXO1 is the driver of the disease strictly required for the survival of tumor cells [5]. PAX3-FOXO1 being a transcription factor (TF), it is considered undruggable even if new approaches to directly blocks TFs are ongoing objects of deep investigations [6]. Embryonal RMS is devoid of any fusion gene (fusion-negative (FN)-RMS) but frequently harbors RAS mutations ([7] and reviewed in [8]). It has been demonstrated that 20–30% of alveolar RMSs that are fusion negative have the same clinical and molecular features of embryonal RMS and are, thus, considered FN-RMS [9]. FN-RMSs have a favorable prognosis when non-metastatic or in advanced stage (5-year survival rates reaching 70–80%) [10]. Overall, survival of RMS patients with localized non-metastatic disease has greatly improved in the last decades due to the multimodal therapeutic approach including chemotherapy, surgery, and, frequently, radiotherapy [11,12]. Conversely, despite heavy multimodal therapeutic regimens, metastatic FN-RMSs and FP-RMSs have a dismal prognosis, showing a 5-year survival rate <30% [10]. Several clinical trials have been developed to identify therapeutic approaches that could improve the prognosis of RMS at a high risk of relapse, which have been supported by two major cooperative groups such as the Children Oncology Group (COG) in North America and the European Pediatric Soft Tissue Sarcoma Study Group (EpSSG) in Europe.

The first-line standard of care for high-risk RMS includes vincristine and dactinomycin combined with an alkylating agent which is cyclophosphamide for COG (VAC) and ifosfamide for EpSSG (IVA) [13,14]. Recently, the EpSSG evaluated the incorporation of doxorubicin to IVA in the phase 3 trial EpSSG RMS 2005. Unfortunately, RMS patients treated with IVA+Doxo did not show any significant improvement in 3-year event-free survival vs. those treated with IVA alone (65.5% and 63.3% in the IVA-Doxo and IVA, respectively) while adverse effects such as leukopenia, infections, and thrombocytopenia were more common than in the IVA group [15]. In addition, continued maintenance chemotherapy with vinorelbine, a semisynthetic analog of vincristine, and low doses of cyclophosphamide was analyzed in high-risk RMS patients in remission after standard treatment in the phase 3 trial EudraCT (NCT00339118; www.clinicaltrials.gov) (follow-up is still ongoing). Five-year disease-free survival (DFS) and overall survival (OS) resulted both higher than those in the group without maintenance therapy (77.6% vs. 69.8% DFS and 86.5% vs. 73.7% OS, respectively) prompting the EpSSG to incorporate this approach in the new standard of care for these types of patients [16].

Of note, patients with bone or bone marrow metastases or metastases in more than 3 sites show a 5-year survival rate <10% (reviewed in [17]). Therefore, one of the major determinants of tumor progression is the presence of metastases at diagnosis or the development of a metastatic disease during therapy, which are both related to the intrinsic marked ability of mesenchymal muscle precursors to migrate and invade during somitogenesis (reviewed in [18]). Indeed, somites are epithelial-condensed structures of mesodermal cells that undergo secondary epithelial-to-mesenchymal transition (EMT), giving rise to highly migrating mesenchymal progenitors committed to skeletal muscle lineage (reviewed in [18]). As for pediatric cancers in general, the inability to differentiate and the migratory properties of RMS cells are related to abnormalities of developmental pathways that regulate skeletal muscle determination [19]. Among these, FN-RMS receptor tyrosine kinase (RTK) signaling involving the RAS/MEK/ERK and the phosphatidylinositol 3-kinase (PI3K)/AKT/mammalian target of rapamycin (mTOR) cascade include upstream and downstream mutated components that in the FP-RMS variant are aberrantly regulated by PAX3-FOXO1, suggesting tumorigenic pathways in the two RMS subtypes converge on a common genetic axis [7]. In the last years, several clinical trials have been started using targeted therapy against molecules involved in deregulated pathways in RMS, which were previously validated in the preclinical setting (Table 1). Results from these clinical trials seem to indicate that therapy with targeted drugs has modest effects on the progression of the disease as single treatment, but should be used in combination with the standard of care as an adjuvant approach [20]. Therefore, novel approaches that should halt tumor metastasis and progression, preventing in the meantime harmful side-effects for young patients affected by RMS, are needed.

Focal adhesion kinase (FAK) is a 125 kDa non-receptor tyrosine kinase encoded by the *PTK2* gene, mainly localized to cellular focal contacts at the cellular edges, which plays a critical role in adhesion-dependent cell motility, survival, and proliferation in response to integrin and RTK signaling [29]. Overall, FAK coordinates signals between the cytoskeleton of the cells and the extracellular microenvironment. These functions make FAK a crucial factor during tissue development, embryogenesis, and cancer by inhibiting cell death after the disruption of adhesions between cells and the extracellular matrix (ECM), i.e., “anoikis” [30,31]. Anoikis is a form of apoptosis that constitutes one of the key defense mechanisms for preventing cancer metastasis [32]. While in most adult tissues FAK is expressed at low levels, in cancer its expression/activation is frequently upregulated and, in certain tumors, negatively correlates with prognosis [33,34]. FAK has been shown upregulated and overactivated in RMS and its inhibition decreases tumor growth in vivo.

In this review, we summarize the insights into the involvement of FAK in RMS pathogenesis. We also discuss the perspectives and challenge of potential clinical applications of FAK inhibitors in RMS.

## 2. FAK Structure and Activity

FAK is not only a sensor of environmental rigidity, but it is also involved in an intricate network of intramolecular interactions existing among the microenvironment, the adhesion receptor complexes, and the nucleus coordinating signals through the focal adhesion multiprotein complex [35]. Functionally, focal adhesion complex works by anchoring the cytoplasmic tails of integrins, which are heterodimeric membrane-spanning proteins, allowing a link with the ECM. This binding provides integrins, lacking for kinase activity, the ability to transduce the signal via FAK in response to changes in cytoskeletal tension.

The structure of FAK consists of multiple domains, including the N-terminal 4.1, ezrin, radixin, moesin homology domain FERM, a central catalytic tyrosine kinase domain important for its activity, and a C-terminal region containing a focal-adhesion targeting (FAT) domain and a proline-rich region [35] (Figure 1). The FERM domain is composed of three lobed structures (lobes F1, F2, F3) arranged in a clover leaf-shaped assembly, and contains a nuclear export sequence (NES) in the lobe F1 and a nuclear localization sequence (NLS) in the lobe F2 [36]. The central kinase domain adopts a typical two-lobed fold. This region contains the activation loop, which extends over 21 residues (564–585) and which is unphosphorylated and highly flexible in the inactive state. It includes two tyrosine residues in the activation loop, Y576 and Y577, which are phosphorylated by the Src family kinases upon activation by cell surface integrins, regulating the kinase activity of FAK. Lastly, the C-terminal domain contains proline-rich regions that consist of two polyproline (PxxP) motifs interacting with the Src homology (SH) 3 domain of several proteins. The extreme C-terminus contains a four-helix bundle that comprises the FAT domain, which interacts with other focal adhesion proteins and is responsible for targeting FAK into the focal adhesion complex [37].

The activation of FAK is under the control of different phosphorylation events. An important autoregulatory mechanism occurs through an interaction between the FERM and kinase domains to maintain an autoinhibited state. A main interaction is formed between the FERM F2 lobe and the kinase C-lobe. In addition, an indirect contact is formed in autoinhibited FAK between the FERM-F1 lobe and the kinase N-lobe. This connection includes the Y397 autophosphorylation site, important for the activation of the protein [37]. FERM has a prominent role in controlling FAK activation as demonstrated by the evidence that the deletion of the first 375 residues of the FERM domain results in constitutive activation of FAK [38]. Moreover, in response to different stimuli, such as receptor tyrosine kinases, intracellular pH changes, integrin recruitment to ECM, G-protein-coupled receptors and cytokine receptors, FAK autoinhibition is removed, thus triggering FAK autophosphorylation at the Y397 site. This process is necessary for FAK activation and recruitment at focal adhesions [39].

Once Y397 autophosphorylation occurs, this site is converted into a high affinity-binding site for proteins containing SH2 and SH3 domains, among which is the Src kinase. In this way, Src interacts via its SH2 domain with pY397 and via the SH3 domain with a PxxP motif located between the FERM and the kinase domain. These interactions contribute to Src-dependent activation, triggering phosphorylation of several FAK tyrosines, including Y576 and Y577 in the activation loop of the kinase domain, which induces full catalytic activity of the protein [37]. Furthermore, an intramolecular interaction has been demonstrated between the FERM domain and the C-terminal FAT domain. This FAT/FERM interaction could promote the recruitment of FAK into focal adhesions, triggering protein dimerization [29]. Once activated, FAK is crucial for the regulation of assembly and disassembly of focal adhesion complexes formed by different types of molecules that perceive the mechanical and biochemical signals from the ECM and transmit them inside the cell, by activating different signal transduction pathways. In fact, FAK has long been known as a regulator of cell migration, but also coordinates several cellular processes including polarization, survival, and proliferation via its kinase activity [39]. FAK activation through ADRB2-dependent activation of Src has been involved in the reduction of anoikis levels in ovarian cancer cells under adrenergic modulation [40]. In liver cancer, the Zinc finger protein 32 (ZNF32) enhances the phosphorylation and activation of Src/FAK signaling, contributing to anoikis resistance [41].

Otherwise, FAK plays its roles via non-canonical kinase-independent scaffold functions. Lim et al. found that FAK was also functional into the nucleus [42]. Indeed, FAK can shuttle between the cytoplasm and the nucleus via NLS in response to stress signals or detachment of cells from the ECM [43]. Nuclear FAK establishes a direct interaction with p53 and Mdm2, thus enhancing Mdm2-dependent p53 ubiquitination, leading to the degradation of p53 through the ubiquitination pathway and inhibiting apoptosis, thus promoting cell survival [42,44] (Figure 1). Nuclear FAK can also synergize with various E3 ligases to promote the turnover of several transcription factors, thus controlling different networks involved in the inflammatory signaling pathway, in the immune escape and in angiogenesis. In this way, FAK influences multiple cell functions, and FAK signaling activation is a hallmark of tumor cells, forming a nodal interconnection among pathways crucial for cancer progression (Figure 2) (reviewed in [45]).

However, the molecular mechanisms are still unclear and further studies on the roles of nuclear FAK are needed [44].

## 3. FAK in Skeletal Muscle

During muscle development, FAK is involved in several processes including differentiation and migration. Crosstalk between myoblasts and ECM, in particular laminin, regulates myogenic processes by stabilizing multinucleated structures that will form muscle fibers (reviewed in [46]. Forced overexpression of a constitutively active membrane-bound FAK in quail myoblasts prompted them to proliferate faster, while overexpression of an inactive FAK mutated in the activation site resulted in differentiation and myotube fusion [47].

This result has been one of the first experimental evidences of an involvement of FAK in myogenesis. A schematic representation of FAK involvement in myogenesis is reported in Figure 3. Interestingly, the activation of FAK under muscle differentiation cues in vitro appears to be biphasic, with a decrease in phosphorylation during the early phases, after the switch of myoblasts from the proliferating to differentiation medium, followed by a gradual increase in cells maintained in differentiation medium for 6 days [48]. This phenomenon is mirrored by treatment with insulin, which stimulates FAK phosphorylation in proliferating myoblasts, followed by a rapid decrease in differentiated cells and again an increase to return to the baseline FAK phosphorylation levels after a half-hour [49]. Moreover, when differentiating myoblasts are transfected with a wild-type FAK, the reduction of the pro-proliferative cyclin D1 is inhibited together with increase of MYOG and muscle creatin kinase levels [48].

Notably, FAK activation is needed for MyoD expression in murine myoblasts and, in addition, FAK cytoplasmic interaction with the methylating protein MBD2 promotes its shuttling to the nucleus and decreases the interactions of MBD2 with the myogenin promoter, thus inducing the expression of the gene (reviewed in [46]). Therefore, FAK activity seems to be strictly related to myogenesis, being downregulated in the early phases to impede proliferation and then present during differentiation, when presumably the function of integrins in regulating myoblasts fusion is mandatory. In agreement with a role in the homeostasis of muscle fibers, FAK is more expressed in myotubes than in myoblasts and involved in muscle regeneration (reviewed in [46]). A crosstalk between FAK and PI3K/AKT/mTOR in muscle cells has been suggested by data showing that IGF1 induces phosphorylation of both FAK and tuberous sclerosis complex 2 (TSC2), which is a negative regulator of mTOR and a target of FAK [50]. This phenomenon leads to the activation of mTOR, which controls protein synthesis and the cell size of the muscle fibers. In agreement, silencing of FAK reduces TSC2 phosphorylation, which is associated with decreased phosphorylation of the mTOR target genes S6K1 kinase and 4E-BP1 [51]. A direct interaction between FAK and PI3K, after binding of FAK to the SH2 domain of the 85 kDa PI3K, increases PI3K activity [52]. Finally, in murine myoblasts, silencing of PTEN or SHP2, two negative regulators of PI3K, results in FAK activity enhancement [53].

Altogether, these data suggest that FAK plays fundamental roles in skeletal muscle participating, in addition to myogenesis/regeneration, in the control of energy metabolism also via PI3K signaling.

## 4. FAK in Rhabdomyosarcoma

The levels of FAK expression cannot be used as a prognostic indicator in RMS since FAK and Y379 phosphorylated FAK are both also expressed in normal skeletal muscle. The expression of FAK in RMS has been evaluated in a recent study showing that, in a small cohort of pediatric RMSs including both alveolar and embryonal histotypes, FAK was expressed and phosphorylated with no difference between the two subtypes [54]. Functionally, a first report has involved FAK in RMS cell motility using a model of inhibition of mTOR signaling (Figure 4) [55]. Of the two mTOR kinase complexes, mTORC1 and mTORC2, both including mTOR but associated with different interactors, only the former phosphorylates the S6K1 kinase and 4E-BP1 to regulate protein biosynthesis, controlling cell growth and size and is sensitive to the macrolide antibiotic rapamycin [56,57]. However, the two complexes are closely related, and they can regulate each other. Then, rapamycin can inhibit mTORC2 too in certain conditions (reviewed in [58]). mTORC1 is downstream to the PI3K pathway and is activated by growth factors (reviewed in [58]). Dysregulated mTOR signaling is a hallmark of many tumors and rapamycin analogs are widely used in clinics and have been investigated in clinical trials (www.clinicaltrials.org) on several adult and pediatric tumor types including RMS ([59] and reviewed in [58]). Activation of mTOR has been reported in RMS [60] and mTORC1 inhibitors have entered clinical trials even if they did not show any significant effect on RMS patients’ prognosis [61]. Liu et al. [55] studied the mechanism of cell motility inhibition of rapamycin on the PAX3-FOXO1 RMS cell line RH30 upon treatment with IGF1. Treatment with the growth factor quickly induced the reorganization of cytoskeleton with F-actin fibers condensed at the leading edge of cells and lamellipodia formation [55]. This phenomenon was inhibited by rapamycin [55]. mTOR was directly involved in the effects of IGF1 as demonstrated by the inability of rapamycin to inhibit cytoskeletal reorganization in the presence of a rapamycin-resistant mutant mTOR (mTORrr). In agreement, genetic silencing of mTOR or the mTOR-associated protein raptor mirrored the rapamycin-dependent effects in IGF1-treated cells. While the levels of focal adhesion components FAK, paxillin, and p130^Cas^ were unmodified by IGF1, the growth factor caused robust phosphorylation of the three proteins, which was prevented by pretreatment with rapamycin which was ineffective in cells expressing mTORrr [55]. IGF1-dependent phosphorylation of FAK and the other focal adhesion proteins was inhibited by the silencing of both raptor and rictor, the latter an mTORC2 component, while rapamycin affected only mTORC1 signaling, suggesting that both complexes participate in this process. Moreover, the effects of rapamycin were abrogated by overexpression of S6K1, the constitutively active effector of mTORC1, and mimicked by S6K1 depletion [55]. Considering that RMS is characterized by high invasiveness and RMS cells exhibit elevated migration properties, these results unveiled the potential of an FAK blockade via pharmacologic mTORC1 inhibition in RMS.

Cell motility can be regulated by FAK as a downstream effector of the MET oncogene [62], which is a regulator of the invasive abilities of cancer cells and overexpressed in both RMS subtypes, in addition to be induced by PAX3-FOXO1 [63,64]. MET is a target of the muscle-specific microRNAs miR-1/miR-206 that drive skeletal muscle differentiation in muscle thus called “MyomiRs” (reviewed in [65]) (Figure 4). In RMS, both these microRNAs are highly downregulated and when forcedly reintroduced in tumor cells are able to halt cell proliferation and migration causing myogenic tumor cell differentiation in vitro and in vivo partly by decreasing MET [2,66]. Yan et al. showed that FAK activation/phosphorylation was impaired after miR-1 and miR-206 overexpression in the FN-RMS cell line, RD, in parallel with MET downregulation and the blockade of cell migration and proliferation [66]. This report suggested an interconnection between the functions of a known oncogene in RMS and FAK activation.

A subsequent study directly involved FAK in cell survival, invasion, and migration of RMS cells, regardless their fusion gene status [54]. As a matter of fact, FAK depletion via siRNAs in FN-RMS RD cells and in PAX3-FOXO1 RH30 cells significantly reduced both cell viability and proliferation. Moreover, the FAK inhibitor PF-573228 [67], which competitively targets the ATP-binding pocket of FAK, blocking autophosphorylation at the Y397 site, also decreased cell survival and cell proliferation, promoting apoptosis via the PARP cleavage and Caspase-3 activation in both cell lines. Further, significant reduction of cell invasion through a Matrigel^®^ layer and cell migration was detected after PF-573228 treatment at different concentrations. The effects of FAK inhibition in vivo were evaluated using Y15, a small molecule that inhibits FAK by decreasing its protein levels [68]. Y15 treatment of RD and RH30 cell lines mirrored in vitro the results obtained with PF-573228, inhibiting cell proliferation, viability, migration, and invasion [54]. When treated with Y15, tumor xenografts, obtained by injecting RD and RH30 cells into immunocompromised mice, grew more slowly and exhibited a smaller mass and lower percentage of the proliferative marker Ki67 compared to those treated with the vehicle [54]. This work suggested for the first time that FAK inhibition could have an impact on RMS tumorigenic features in vivo, paving the way to preclinical investigation on inhibition of this kinase in this soft tissue tumor.

The Y397 phosphorylated FAK was detected in the FN-RMS cell line RD18, a subclone of RD cells [69], by mass spectrometry (MS)-based phosphoproteomics [70]. Recently, Narendran et al. showed that in FN-RMS RD cells, FAK autophosphorylation at Y379 appeared higher, while FAK Y576/577 phosphorylation levels at the activation loop were lower than those in the hTERT-immortalized normal primary fibroblasts [71]. Conversely, no difference in the levels of expression was found in the phosphorylation of Y925 FAK between the two cell types [71]. Treatment with the ATP-competitive, reversible inhibitor of FAK, PF-562271, markedly reduced Y397 phosphorylation of FAK in RD cells and suppressed the co-localization of FAK Y397 phosphorylation with F-actin stress fibers and the cell edges, indicative of loss of the motile cell phenotype [71]. FAK inhibition also determined a significant cell cycle arrest in the G1 phase of tumor cells compared to the vehicle. Then, the authors explored the migratory phenotype of RD cells, showing that PF-562271 treatment prevented cell migration, as reported by Waters et al. [54] for PF-573228. These findings directly linked FAK activation with cell motility and F-actin fibers’ organization in FN-RMS cells [71].

Interestingly, the alkylating agent Temozolomide has been shown to impair RH30 FP-RMS cells’ survival and, at the same time, to activate autophagy, the inhibition of which synergizes with the drug to induce tumor cell death [72]. Through autophagy, tumor cells can enzymatically degrade and recycle cytosolic components under stressful conditions such as the extracellular matrix (ECM) detachment and starvation [73,74]. Autophagy is a mechanism through which anoikis can be inhibited and, thus, involved in anoikis resistance in solid tumors [75]. FAK being involved in the inhibition of autophagy by phosphorylating Beclin-1, a component of the complex promoting autophagosome biogenesis [76], and by activating mTOR [39], the involvement of FAK in the response to Temozolomide in FP-RMS cells deserves further investigations.

Recently, Fanzani et al. demonstrated that caveolin-1, when overexpressed in FN-RMS cells in vitro, promotes migration and invasion of tumor cells, a phenomenon blocked by ERK inhibition [77]. Caveolin-1 is one of the major Src kinase substrates that localizes to the caveolae and to focal adhesions sites to drive directional motility and, when phosphorylated/activated by Src, stabilizes FAK, stimulating cell motility in prostate cancer cells [78]. Moreover, caveolin-1 is a target of mTORC2 that, through caveolin-1 phosphorylation, regulates the formation of caveolae [79]. Since mTORC2 cooperates with FAK in mesenchymal stem cells [80] and caveolin-1 stabilizes FAK [78], these pathways could also be interconnected in RMS, where all these signalings are deregulated and warrant future investigations.

## 5. Future Perspectives

The multiple roles of FAK make this protein crucial for oncogenesis and, in agreement, FAK is overexpressed in most types of cancer [81,82]. The blockade of FAK phosphorylation at Y397 decreases several processes needed for tumorigenesis such as migration, invasion, and survival [83,84]. Changes in FAK expression and phosphorylation have been found to correlate with many different types of adult cancers, including hepatocellular carcinoma, breast, colon, brain, and ovarian cancers [85,86]. Notably, there is also an evident correlation between overexpression levels of FAK and worst prognosis in several types of human cancers in the adult setting, including those of the colon, thyroid, and liver [33,34]. Considering that those tumors in which FAK is overexpressed could benefit from strategies aimed to lower its protein levels [87], a discrete number of studies on FAK inhibition have been carried out to investigate the anti-tumor effects of FAK depletion in cancer cells so far [88]. Moreover, small compounds have been discovered to inhibit FAK signals. Most FAK inhibitors are designed to be competitive for the ATP-binding site, which is located in the kinase domain of FAK [85], or to be scaffold inhibitors, binding the hinge region of the kinase domain of the protein, thus blocking the access of ATP to the ATP-binding site (reviewed in [89]). Among those inhibitors, some drugs are currently under evaluation in phase I and II clinical trials or in observational studies in patients with solid tumors (reviewed in [89]) (Table 2).

Since FAK has several interactors, a huge effort is in place to discover protein–protein interaction (PPI) inhibitors of FAK as a strategy to avoid issues of selectivity related to competitive inhibitors of the ATP-binding site. Among the FERM interaction inhibitors, roslin 2 is a compound that, blocking the p53-binding site in the F1 lobe of FERM, reduces FAK levels and reactivates p53 in colon cancer cells, synergizing with conventional chemotherapeutics [90].

With the same principle, the small molecule M13 blocked interactions between the FERM domain of FAK and MDM2, leading to an increase in p53 levels and a decrease of tumor cells’ survival [91].

The compound INT2-31 targets the FAK/IGF1-R interaction and has been used in preclinical research on melanoma, showing the ability to block cell proliferation inducing apoptosis in vitro and decreasing tumor growth in vivo by inhibition, at least in part, of AKT signaling [92].

An in silico screening to target the Y397 site identified two small molecules, Y11 and Y15, which inhibit FAK autophosphorylation, reducing the levels of the protein, thus targeting the FAK binding to SH2-containing proteins [93,94]. Y11 affected in vivo colon cancer tumor growth [93] and Y15 synergized with the antimetabolite and pyrimidine analogue 5-fluorouracil (5-FU) and Temozolomide in colon cancer and glioblastoma xenografts models with no toxic effects at doses needed for FAK inhibition [94].

One of the therapeutic advances in which FAK inhibition can be exploited is chemoresistance, often developed by some tumors following the first-line interventions. Several preclinical studies highlighted the anti-tumor effects of an anti-FAK approach in combination with conventional or targeted therapeutics promoting tumor remission in xenografts models (reviewed in [95]). As an example, the BRAF inhibitor vemurafenib, used in patients with melanoma and colon cancer carrying BRAF mutations, hyperactivates FAK with an unknown mechanism, leading to Wnt/β-catenin pathway upregulation in colon cancer [96]. This effect was overcome by the combinatorial use of the competitive FAK inhibitor PF-562271, resulting in tumor growth inhibition in vivo [96]. Recently, an immunotherapeutic approach using PD1 and CTLA4 antibodies in an in vivo model of pancreatic cancer benefited from the combination with the competitive FAK kinase inhibitor BI 853520, leading to the impairment of xenografts’ growth and an increase of mice survival [97]. The preclinical studies on the efficacy of FAK inhibition in combination with other drugs/small molecules in some tumors has paved the way for several clinical trials using this approach in combination with other targeted therapies, chemotherapy, and/or immunotherapy agents (Table 2).

What emerged from the preliminary data obtained from those trials is that FAK inhibitors could be potential cytostatic drugs, capable of keeping the disease stable for more than 12 weeks (reviewed in [89]). These data support the ongoing of research for the discovery and development of new FAK inhibitors to be tested against various types of cancer including pediatric tumors. Preclinical investigations with targeted molecules against pathways that cross-talk with FAK signaling have been done in RMS but they have not yet entered the clinical setting (Table 1). However, a number of clinical trials using small compounds against specific molecules that are directly or indirectly involved in FAK signaling are ongoing or have been recently completed (Table 1). They could be exploited in preclinical studies to evaluate the potential of FAK co-inhibition. It is indeed conceivable that inhibiting FAK in combination with conventional chemotherapy or with compounds targeting FAK-related pathways that are dysregulated in the tumor cells, such as IGF1-R or MEK, deserves further studies also in RMS.

## 6. Conclusions

Future studies should identify the best combinations using FAK inhibitors and small molecules against signaling cross-talking with FAK that could have potentiality in RMS patients. They could be those affecting key pathways involved in the developmental signals that are fundamental for the survival, motility, and stemness of RMS cells. Among these, inhibitors of IGF1-R, as already mentioned, as well as CDK4/6 or PARP inhibitors could be investigated in a preclinical RMS setting. Moreover, a combination strategy could allow for the reduction of the doses for each compound that in therapy with a single agent are higher, thus lowering the side effects often related to targeted therapies.

However, more studies are needed to understand how FAK regulates tumor survival and the exchanges with the tumor microenvironment in cancer and, in particular, in RMS. In addition, the nuclear role of FAK in gene expression should be better clarified. Preclinical models of pediatric cancers can help in elucidating the role of FAK and its signaling in tumor progression and in validating FAK inhibition in combinatorial therapeutic strategies.

## Figures and Tables

**Figure 1 ijms-21-08422-f001:**
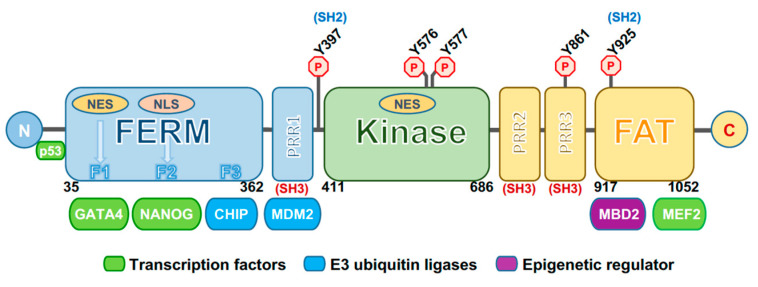
Schematic representation of focal adhesion kinase (FAK). The three main domains of FAK are depicted: The N-terminal 4.1, ezrin, radixin, moesin homology domain (FERM) (in blue), central kinase domain (in green), and focal-adhesion targeting (FAT) domain (in yellow). Domain boundaries are shown. F1, F2, and F3 represent the three lobes of the FERM domain and the regions for binding with transcription factors (such as GATA4 and NANOG) and E3 ubiquitin ligase factors (such as CHIP). Other binding sites are those for MDM2 (E3 ubiquitin ligase), MBD2 (methyl-CpG binding domain protein 2, an epigenetic factor) and MEF2 (transcription factor). Outside the FERM domain the p53-binding site is shown. A nuclear localization sequence (NLS) is located at the F2 FERM lobe, while two nuclear export sequences (NES) are at the F1 FERM lobe and in the kinase domain. PRR1, 2, and 3 represent proline-rich regions (i.e., two polyproline (PxxP) motifs) that interact with the Src homology SH3 domains of several proteins, including Src. Several phosphorylation sites are depicted, among which include Y397, the autophosphorylation site, and Y925, both binding sites for Src. N: N-terminus. C: C-terminus.

**Figure 2 ijms-21-08422-f002:**
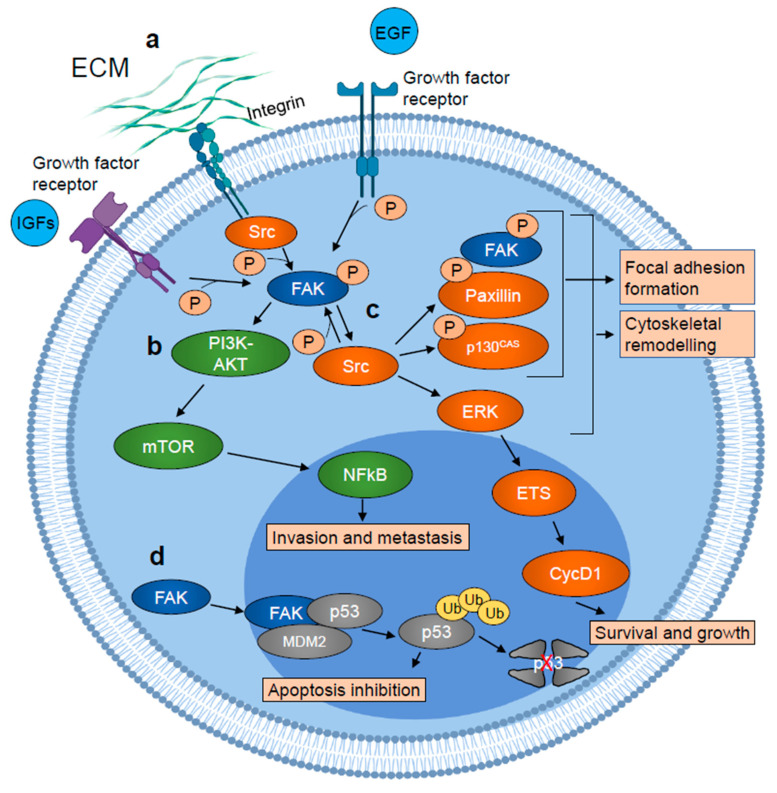
Schematic representation of FAK involvement in tumor growth and metastasis. (**a**) FAK is autophosphorylated in response to growth factor receptors and integrins activation and activated by Src. (**b**) Active FAK promotes tumor cell invasion and metastasis, activating PI3K-AKT-mTOR signaling cascade, which results in increased NFkB transcriptional activity. (**c**) Active FAK also stimulates cytoskeletal remodeling and focal adhesion formation/turnover, inducing SRC-dependent phosphorylation of paxillin and p130cas, leading to the formation of a focal adhesion complex which includes phosphorylated/active FAK, paxillin, and p130cas. SRC also stimulates ERK signaling cascade which results in the ETS transcription factor-dependent induction of cyclin D1 (CycD1) expression which in turn promotes tumor cell survival and growth. (**d**) Nuclear FAK acts as a scaffold protein for the p53–MDM2 interaction, inducing p53 ubiquitination and its proteasomal degradation which results in apoptosis inhibition. Figure realized with BioRender.com.

**Figure 3 ijms-21-08422-f003:**
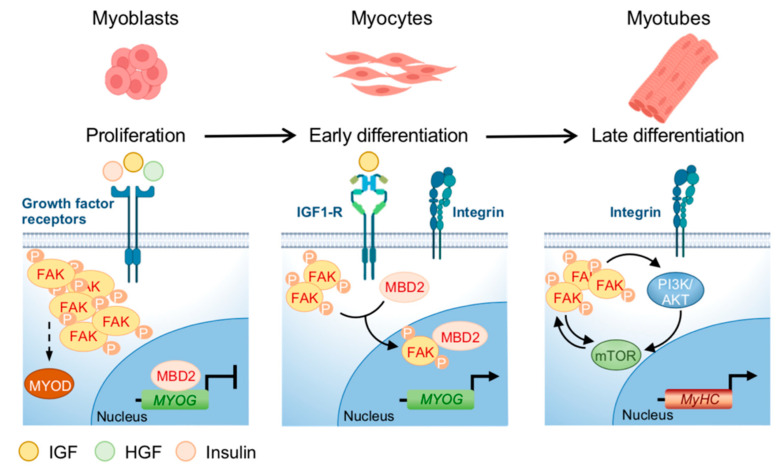
FAK regulation during myoblasts differentiation. The expression of FAK and its phosphorylation are related to specific points in the differentiation of myoblasts into myotubes. FAK is activated by growth factor receptors and regulates the development of myoblasts and the formation of muscle fibers. During the proliferation phase, the activation of growth factor receptors leads to FAK phosphorylation/activation, needed for MYOD expression, while the expression of MYOG is blocked by the binding of MBD2 to its promoter. In the early differentiation phase, phosphorylated-FAK fraction decreases and FAK cytoplasmic interaction with MBD2 promotes the translocation of the FAK/MBD2 complex into the nucleus, where MBD2 interaction with the MYOG promoter is prevented, leading to MYOG expression. During myotubes formation, FAK levels increase, even if not at the same level as in the proliferative phase. In this terminal phase, characterized by *MyHC* expression, integrins activation induces multiple pathways such as PI3K/AKT/mTOR which crosstalk with FAK to control protein synthesis and the size of the muscle fibers. Figure realized with BioRender.com.

**Figure 4 ijms-21-08422-f004:**
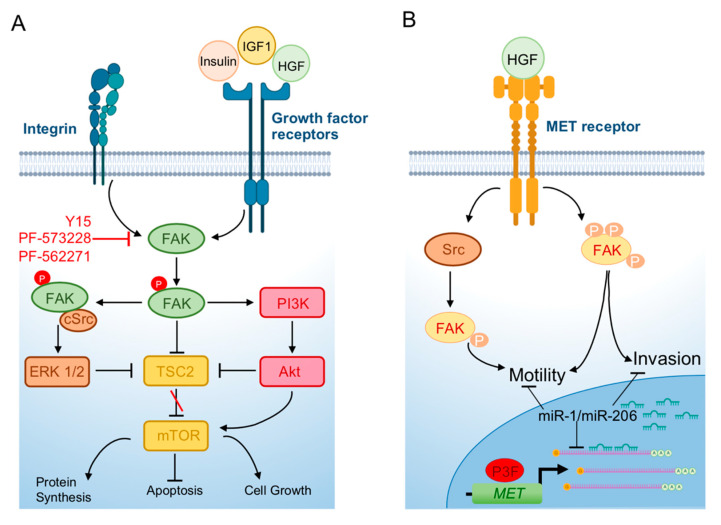
FAK involvement in rhabdomyosarcoma. (**A**) FAK is phosphorylated and activated upon binding of growth factors to the RTK receptors and integrins activation. Phosphorylated FAK promotes cell growth and protein synthesis and blocks apoptosis through activation of PI3K-AKT-mTOR signaling. The FAK–Src complex (i) induces the ERK1/2 pathway, leading to the reduction of phosphorylated TSC2 which results in increased mTOR activity; and (ii) blocks TSC2 directly. (**B**) In rhabdomyosarcoma, MET, which is a transcriptional target of PAX3-FOXO1 (P3F), is overexpressed and induces signal cascades, promoting motility and invasion also through FAK activation. In these tumor cells, the “MyomiRs” miR-1/miR-206 are downregulated and when forcedly expressed, they target MET mRNA and block its translation, resulting in a block of invasion and motility. Figure realized with BioRender.com.

**Table 1 ijms-21-08422-t001:** Preclinical and clinical trials with targeted therapy in rhabdomyosarcoma (RMS).

Molecular Target	Drug	Phase	Clinicaltrials.Gov Identifier (USA)	References
GSK3β	Tideglusib, LY2090314, 9-ING-41	Preclinical	NA	[21,22]
NOTCH	RO4929097, MK0572, brontictuzumab (mAb), tarextumab (mAb)	Preclinical	NA	[23,24]
Bcl-2	Venetoclax (ABT-199), ABT-737	Preclinical	NA	[25,26]
PLK1	Volasertib	Preclinical	NA	[27]
XIAP	SMAC mimetics (LCL161)	Preclinical	NA	[28]
**Molecular Target**	**Drug**	**Phase**	**Clinicaltrials.gov Identifier (USA)**	**Primary Purpose**
BET	BMS-986158	Clinical (I)	NCT03936465	Treatment
ALK	Crizotinib	Clinical (II)	NCT01524926 **	Treatment
**PI3K/mTOR**	**Temsirolimus**	**Clinical (II)**	**NCT00106353 NCT01222715**	**Treatment** **Treatment in combination**
MEK1	Cobimetinib	Clinical (I)	NCT04216953	Treatment in combination
FGFR	Erdafitinib	Clinical (II)	NCT03210714 NCT03155620	Treatment Screening drugs
**IGF-1R**	**R1507 (mAb)**	**Clinical (II)**	**NCT00642941**	**Treatment**
**VEGF**	**Bevacizumab (mAb)**	**Clinical (II)**	**NCT01222715**	**Treatment in combination**
Multi-RTKs	Regorafenib	Clinical (II)	NCT01900743 *	Treatment
**SMO**	**LDE225**	**Clinical (I)**	**NCT01125800**	**Treatment**
CDK4/6	Palbociclib	Clinical (I) Clinical (II) Clinical (II)	NCT03709680 NCT03526250 NCT03155620	Treatment in combination Treatment Screening drugs
CDK4/6	Abemaciclib	Clinical (I) Clinical (I)	NCT02644460 NCT04238819	Treatment Treatment in combination
Wee1	AZD1775	Clinical (II)	NCT02095132	Treatment in combination
PARP	Olaparib	Clinical (II)	NCT03155620 NCT03233204	Treatment Treatment

* Soft tissue sarcoma; ** alveolar rhabdomyosarcoma; in **bold**, completed clinical trials.

**Table 2 ijms-21-08422-t002:** Summary of completed, terminated, and active clinical trials with FAK inhibitors.

Drug (Code Name), Trade Name	Target	Clinical Trial Studies (a, b)	No. of Clinical Trials (a)	Phase (a)	Interventions (a)
APG-2449	FAK, ALK, ROS1	Esophageal cancer, mesotelioma, NSCLC, ovarian cancer, solid cancer	1; 1 active	Phase I: 1	1; 1 single agent
BI-853520 (IN-10018)	FAK	Advanced or metastatic solid tumors, metastatic melanoma	3; 2 completed/terminated, 1 active	Phase I: 3	3; 3 single agent, 1 combination
Dasatinib (BMS-354825) SPRYCEL	Abl, Src, c-Kit and FAK	ALL, AML, bladder carcinoma, bone metastases, breast carcinoma, CNS, cervical carcinoma, chondrosarcoma, CLL, CML, CRC, epithelioid sarcoma, esophageal carcinoma, fallopian tube cancer, GIST, glioblastoma, lioma, head and neck carcinoma, hepatocellular carcinoma, Hodgkin’s lymphoma, kidney carcinoma, lung Carcinoma, lymphoma, lymphoma, non-Hodgkin, liver carcinoma, mesothelioma, melanoma, MDS, myeloma, neuroblastoma, NSCLC, ovarian carcinoma, PDAC, prostate carcinoma, rhabdomyosarcoma, sarcoma, Ewing’s sarcoma, skin carcinoma, solid neoplasm, solid tumor, SCS, testicular germ cell tumor, thyroid gland carcinoma, tongue cancer, urinary bladder neoplasms, urothelial carcinoma, uterine corpus cancer	279; 181 completed/terminated, 98 active Of which 45 pediatric; 22 completed/terminated, 23 active	Early Phase I: 2 Phase I: 62 Phase I/II: 28 Phase II: 130 Phase II/III: 3 Phase III: 16	279; 94 single agent, 185 combination
Defactinib (PF-04554878, VS-6063)	FAK	bladder carcinoma, breast carcinoma, cervical carcinoma, CRC, endometrial carcinoma, esophageal carcinoma, gastric carcinoma, glioma, head and neck carcinoma, hematopoietic neoplasm, kidney carcinoma, liver carcinoma, lung carcinoma, lymphoma, melanoma, mesothelioma, myeloma, NSCLC, ovarian cancer, PDAC, prostate carcinoma, skin carcinoma, thyroid gland carcinoma, uterine corpus cancer	16; 7 completed/terminated, 9 active	Phase I: 7 Phase I/II: 2 Phase II: 7	16; 5 single agent, 11 combination
GSK-2256098	FAK	Cancer, meningioma, pancreatic cancer	4; 3 completed/terminated, 1 active	Phase I: 3 Phase II: 2	4; 2 single agent, 2 combination
PF-00562271 (PF-562271)	FAK	Head and neck cancer, pancreatic cancer, prostatic cancer	1; 1 completed/terminated	Phase I: 1	1; 1 single agent
PND-1186 (VS-4718, SR-2156)	FAK	AML, ALL, metastatic cancer, pancreatic cancer	2; 2 completed/terminated	Phase I: 2	2; 1 single agent, 1 combination

(a): From www.clinicaltrials.gov accessed 29 Settember 2020. (b): ALL, acute lymphocytic leukemia; AML, acute myeloid leukemia; CLL, chronic lymphocytic leukemia; CML, chronic myeloid leukemia; CNS tumor, central nervous system tumor; CRC, colorectal cancer; GIST, gastrointestinal stromal tumors; MDS, myelodysplastic syndromes; NSCLC, non-small-cell lung carcinoma; PDAC, pancreatic ductal adenocarcinoma; SCC, squamous cell carcinoma; SCS, spindle cell non-osteogenic bone sarcomas. ALL, acute lymphocytic leukemia; AML, acute myeloid leukemia; CLL, chronic lymphocytic leukemia; CML, chronic myeloid leukemia; CNS tumor, central nervous system tumor; CRC, colorectal cancer; GIST, gastrointestinal stromal tumors; MDS, myelodysplastic syndromes; NSCLC, non-small-cell lung carcinoma; PDAC, pancreatic ductal adenocarcinoma; SCC, squamous cell carcinoma; SCS, spindle cell non-osteogenic bone sarcomas.

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
