# Peer review of "FAK Signaling in Rhabdomyosarcoma"

_ijms, 2020, doi:10.3390/ijms21228422_

Round 1
Reviewer 1 Report
- Abstract should also include 2-3 summary sentences on the role of FAK in RMS (aRMS, eRMS, pleomorphic RMS, spindle cell/sclerosing)
- Mention about the current standard of care (surgery, multimodal therapy: chemo (Bisogno Lancet Oncol 2018: 19: 1061-71), surgery, radiation) in localized and metastatic settings and the responses achieved with those treatment.
- Mention about ongoing clinical trial with targeted therapy in RMS according molecular alterations found in RMS for instance ALK rearrangement (ALK in aRMS EORTC 90101 CREATE: Schoffski et al EJC 2018)
- If possible/available, can they tabulate the available preclinical/ clinical (future) trials in RMS (which RMS subtype?) that utilizes FAK pathway.
-Figure 1: FAK structure. FERM domain not FARM domain.
-Part 3 (FAK in skeletal muscle) and part 4 (FAK in RMS). FAK pathway in skeletal muscle and RMS could be summarize with a scheme. FAK pathway crosstalk with PI3K/AKT/mTOR as well as MET, PAX3-FOXO1 found in RMS should be highlighted.
-Table 1: remove SCLC in abbreviation section. It doesn’t appear in the table.
Author Response
Response to Reviewer #1:
-Abstract should also include 2-3 summary sentences on the role of FAK in RMS (aRMS, eRMS, pleomorphic RMS, spindle cell/sclerosing)
We have included a sentence on the role of FAK in RMS (lines 30-32). We were unable to find any data on FAK in pleomorphic and spindle cell/sclerosing RMS in pubmed. Therefore, also due to space constriction (no more than 200 words for Abstract) we preferred to focus on the argument of the review.
-Mention about the current standard of care (surgery, multimodal therapy: chemo (Bisogno Lancet Oncol 2018: 19: 1061-71), surgery, radiation) in localized and metastatic settings and the responses achieved with those treatment.
As requested in the Introduction section we have added sentences on the current standard of care (IVA) in localized (lines 54-56) and high-risk RMS in combination or not with doxorubicin (Ref.15, Bisogno et al., Lancet Oncol 2018) (lines 58-68). In addition, we have also reported the results of a clinical trial on continue maintenance chemotherapy which will be the new standard of care for patients with high-risk RMS in future EpSSG clinical trials (Ref.16, Bisogno et al., Lancet Oncol 2019) (lines 70-75).
-Mention about ongoing clinical trial with targeted therapy in RMS according molecular alterations found in RMS for instance ALK rearrangement (ALK in aRMS EORTC 90101 CREATE: Schoffski et al EJC 2018)
We have added a new table, named Table 1 (while the old Table 1 has been renamed Table 2), in which we report the clinical trials with targeted therapy in RMS including those completed. Moreover, we added two sentences on these clinical trials and the suggestions given by their results and included the reference of the EORTC 90101 CREATE study (Ref.20) in the Introduction section (lines 90-94).
-If possible/available, can they tabulate the available preclinical/ clinical (future) trials in RMS (which RMS subtype?) that utilizes FAK pathway.
We have added some sentences at the end of “5. Future Perspectives” section reporting preclinical and clinical trials in RMS with targeted therapies that potentially utilizes FAK signaling (lines 404-409).
-Figure 1: FAK structure. FERM domain not FARM domain.
We apologize for the typo error and have corrected this on the figure.
-Part 3 (FAK in skeletal muscle) and part 4 (FAK in RMS). FAK pathway in skeletal muscle and RMS could be summarize with a scheme. FAK pathway crosstalk with PI3K/AKT/mTOR as well as MET, PAX3-FOXO1 found in RMS should be highlighted.
We have added Figure 3 (indicated in line 206) as a schematic representation of FAK levels in myoblasts before myogenesis and in the early and late phase of the process.
We have also added Figure 4 highlighting the connection between FAK and PI3K/mTOR and between FAK and MET/miR-1/miR-206 in RMS.
-Table 1: remove SCLC in abbreviation section. It doesn’t appear in the table.
Thank you so much for pointing to this error, we have also added the spelling of the SCS abbreviation as “Spindle cell non-osteogenic bone sarcomas” that was lost on the previous Table 1, now renamed Table 2.
Please see the attachment

Reviewer 2 Report
This is a good overview of the current knowledge on FAK signaling in rhabdomyosarcoma. The review is well-written, and focuses on structure of FAK protein, its role in normal development and RMS, with a conclusive statement on the need of future research for this cancer target.
Since the expression of FAK cannot be used as a prognostic indicator in RMS due to its expression in normal tissues, and it is known to be overexpressed in many adult cancers, there is a clear need to identify its specific prognostic function in RMS.
The combination drug treatments should be explored to understand the role and mechanism of FAK regulation of cell survival, as proposed by the authors. Maybe a brief discussion of best combinations with strong rationale (based on the ongoing studies and trials) can be added to the Conclusions.
The review would benefit from a Figure showing where the FAK protein is currently placed in the network of related cellular pathways, relevant to cancer progression.
Technical error: FERM protein is shown as FARM in Figure 1.
Author Response
Response to Reviewer #2:
Maybe a brief discussion of best combinations with strong rationale (based on the ongoing studies and trials) can be added to the Conclusions.
We have added some sentences in the “6. Conclusion” section about the potentiality of FAKi combinations in RMS based on the ongoing clinical trials. (lines 414-420)
The review would benefit from a Figure showing where the FAK protein is currently placed in the network of related cellular pathways, relevant to cancer progression.
As suggested we have added Figure 2 as a schematic representation of the major pathways in which FAK is involved in cancer for the regulation of metastasis and survival/proliferation of tumor cells adding a sentences in lines 181-183 of “2. FAK structure and activity” section.
Technical error: FERM protein is shown as FARM in Figure 1.
We apologize for the typo error and have corrected this on the figure.
Please see the attachment
